# Particular Anatomy of the Hyperopic Eye and Potential Clinical Implications

**DOI:** 10.3390/medicina59091660

**Published:** 2023-09-14

**Authors:** Maria-Cristina Marinescu, Dana-Margareta-Cornelia Dascalescu, Mihaela-Monica Constantin, Valeria Coviltir, Vasile Potop, Dan Stanila, Farah Constantin, Cristina Alexandrescu, Radu-Constantin Ciuluvica, Liliana-Mary Voinea

**Affiliations:** 1Carol Davila University of Medicine and Pharmacy, 050474 Bucharest, Romania; 2Department of Ophthalmology, Clinical Hospital for Ophthalmological Emergencies, 010464 Bucharest, Romania; 3Department of Ophthalmology, Oftaclinic, 040254 Bucharest, Romania; 4Department of Ophthalmology, Faculty of Medicine, Lucian Blaga University, 550169 Sibiu, Romania; 5Department of Ophthalmology, Faculty of Medicine, Ovidius University, 900470 Constanta, Romania; 6Department of Ophthalmology, Bucharest Emergency University Hospital, 050098 Bucharest, Romania

**Keywords:** hyperopia, emmetropia, corneal hysteresis, corneal resistance factor, endothelial cell density, endothelial cell variability, anterior chamber depth, axial length

## Abstract

*Background and Objectives*: Hyperopia is a refractive error which affects cognitive and social development if uncorrected and raises the risk of primary angle-closure glaucoma (PACG). *Materials and Methods*: The study included only the right eye—40 hyperopic eyes in the study group (spherical equivalent (SE) under pharmacological cycloplegia over 0.50 D), 34 emmetropic eyes in the control group (SE between −0.50 D and +0.50 D). A complete ophthalmological evaluation was performed, including autorefractometry to measure SE, and additionally we performed Ocular Response Analyser: Corneal Hysteresis (CH), Corneal Resistance Factor (CRF); specular microscopy: Endothelial cell density (CD), Cell variability (CV), Hexagonality (Hex), Aladdin biometry: Anterior Chamber Depth (ACD), Axial Length (AL), Central Corneal Thickness (CCT). IBM SPSS 26 was used for statistical analysis. *Results*: The mean age of the entire cohort was 22.93 years (SD ± 12.069), 66.22% being female and 33.78% male. The hyperopic eyes had significantly lower AL, ACD, higher SE, CH, CRF. In the hyperopia group, there are significant, negative correlations between CH and AL (r −0.335), CRF and AL (r −0.334), SE–AL (r −0.593), ACD and CV (r −0.528), CV and CRF (r −0.438), CH (r −0.379), and positive correlations between CCT and CH (r 0.393) or CRF (r 0.435), CD and ACD (r 0.509) or CH (0.384). Age is significantly, negatively correlated with ACD (r −0.447), CH (r −0.544), CRF (r −0.539), CD (r −0.546) and positively with CV (r 0.470). *Conclusions*: Our study suggests a particular biomechanical behavior of the cornea in hyperopia, in relation with morphological and endothelial parameters. Moreover, the negative correlation between age and ACD suggests a shallower anterior chamber as patients age, increasing the risk for PACG.

## 1. Introduction

Hyperopia is one of the most frequent refractive errors, both in the pediatric and adult populations, with an important potential for impact on the daily quality of life [1]. It is estimated that the worldwide prevalence of hyperopia is 4.6% in children and 30.9% in adults, with large variations between different geographic regions [2].

While common, uncorrected hyperopia, and particularly anisometropia (difference in refractive error between the two eyes), raise an important risk for amblyopia (also known as lazy eye) during childhood, as evidenced by a recent study performed on a Romanian pediatric population [3]. Persistent amblyopia has been found to be associated with a poorer self-rated overall health, and to have an impact on mental health and overall well-being [4].

In adults, hyperopia is a known risk factor for primary angle-closure glaucoma (PACG)—an SE between 1.01 and 3.00 Diopters (D) associates an odds ratio of PACG of 1.58, while hyperopia over 3 D associates an odds ratio of 3.33, these figures being even higher in patients younger than 65 years old [5]. Regarding glaucoma, along with the high intraocular pressure (IOP), another important risk factor is represented by biomechanical corneal properties. In PACG, corneal hysteresis has been frequently described as lower compared with healthy controls, even adjusting for age and IOP, and improving after treatment [6]. 

Recent literature has shown significant anatomical differences in the hyperopic eye—such as a higher choroidal thickness in children, which correlates with the axial length [7]. The objective of this study is to better describe the morphological, biomechanical and endothelial properties of the hyperopic cornea, in relationship with axial length and anterior chamber depth, and to compare those to a control group of emmetropic eyes.

## 2. Materials and Methods

This study has a prospective, non-randomized cross-sectional methodology. The study cohort was formed by applying inclusion and exclusion criteria to all patients who consecutively presented to the Oftaclinic Ophthalmology practice, in Bucharest, Romania, between February 2023 and June 2023. The study was conducted in accordance with the Declaration of Helsinki, and approved by the Research Ethics Committee of Carol Davila University of Medicine and Pharmacy (protocol code PO-35-F-03/16.01.2023). Informed consent was obtained from all subjects involved in the study, and from legal guardians in the case of participants under the age of 18. 

The inclusion criteria were:-For the study group: diagnosis of hyperopia (spherical equivalent (SE) over 0.50 D) [8];-For the control group: diagnosis of emmetropia (SE between −0.50 D and +0.50 D) [9].

Patients were included in the study and control groups according to the value of the spherical equivalent calculated after pharmacological cycloplegia (cyclopentolate 10 mg/mL, instilled 3 times every 5 min in both eyes). Furthermore, patients were included in the pediatric group (age under or equal to 18 years old) and the adult group (age over 18 years old).

The exclusion criteria were represented by the presence of ocular pathology, other than hyperopia (myopia, keratoconus, amblyopia, cataract, glaucoma, vitreoretinal pathology), the diagnosis of presbyopia or a history of refractive surgery. Furthermore, patients were excluded in the absence of testing compliance (such as low waveform in Ocular Response Analyser testing, under 7), if the patient was pregnant or if they disclosed any systemic pathology (diabetes mellitus, arterial hypertension, dyslipidemia) or systemic chronic medication. Randomly the right eye of each patient was included in the analysis.

All patients underwent a complete ophthalmological evaluation, including autorefractometry, before and after pharmacological cycloplegia—Topcon KR800 (Topcon, Tokyo, Japan) to measure the spherical error and equivalent (spherical error + 1/2 cylindrical error), slit lamp examination of the anterior and posterior ocular segments, Goldmann tonometry. Additionally, patients underwent the following measurements, also under pharmacological cycloplegia:-Ocular Response Analyzer (ORA) (Reichert Ophthalmic Instruments Inc, Depew, NY, USA)—in order to determine corneal biomechanical properties: corneal hysteresis (CH), corneal resistance factor (CRF) and the Goldmann-correlated intraocular pressure IOP (IOPg);-Aladdin biometer (Topcon, Tokyo, Japan)—to determine the axial length (AL), anterior chamber depth (ACD) and central corneal thickness (CCT);-Specular microscopy (Nidek, Gamagori, Japan) in order to determine corneal endothelial parameters: cell density (CD), coefficient of variation of cell area (CV), percent hexagonality (Hex).

The Ocular Response Analyzer is a non-contact tonometry device applying an air pulse on the corneal surface and following the corneal deformation and its return to the initial state using infrared light. The device records 2 applanation pressures, therefore measuring intraocular pressure and two estimates of corneal viscoelasticity: CH, which represents corneal capacity to absorb and dissipate energy (equal to the pressure difference between the first and second applanation) and CRF, which reflects the global corneal resistance (similar to CH, the second applanation multiplied with a constant) [10]. The measurements with the highest Waveform score were included in the analysis.

The specular microscopy uses the principle of specular light reflection, in which the endothelial layer acts as a mirror, transmitting an image of itself to the device, which analyzes its properties [11].

The Aladdin biometer is an optical low-coherence interferometer measuring ocular morphological parameters [12], and autorefractometry measures refractive errors following the principle of retinoscopy (registering the movement of the retinal reflection of a light, projected towards the eye) [13]. 

### Statistical Analysis

This study includes both categorical and numerical, continuous data. The absolute and relative frequency were calculated for categorical data. For numerical data, the average and standard deviation were determined. 

Levene’s Test, followed by the *t* Test, was applied in order to identify significant differences between the groups (hyperopic and emmetropic, male and female, children and adults). Pearson’s correlation coefficient (“Pearson’s r”) was calculated to determine the degree of correlation between variables. A weak correlation has Pearson’s r between 0.3 and −0.3, a moderate correlation between 0.3 and 0.5 or between −0.3 and −0.5, and a strong correlation over 0.5 or under −0.5. As IOPg may act as a confounding variable, correlations were calculated controlling for it. The *p* value of 0.05 is considered a threshold for statistical significance. The Statistical Package IBM SPSS Statistics for Windows, version 26 (IBM Corp., Armonk, NY, USA) was used to perform the statistical analysis.

## 3. Results

The hyperopic, study group was comprised of 40 eyes (67.50% female, 32.50% male; 45.00% adults, 55.00% children), and the emmetropic, control group was comprised of 34 eyes (64.71% female, 35.29% male; 61.76% adults, 38.24% children). Average age in the hyperopic group was 19.83 years old (age range 6–38), and in the emmetropic group was 26.59 (age range 10–40) (see Table 1).

The hyperopic eyes had significantly lower AL, ACD, higher SE, CH, CRF, and were from significantly younger patients (see Table 1). There were no significant differences between males and females, in the entire cohort, study or control group.

There are statistically significant differences between adults and children: lower ACD, CH, CRF and CD in adult hyperopes and emmetropes, and significantly higher CV in adult hyperopes compared to pediatric hyperopes (See Table 2).

In the hyperopia group, there are several significant correlations between variables—all statistically significant ones can be found in Table 3. Thus, in the hyperopia group there are significant strong negative correlations between Age-CH, Age-CRF, AL-SE, ACD-CV, Age-CD, moderate negative correlations between Age-ACD, AL-CH, AL-CRF, Hex-CV, CH-CV, CRF-CV, CD-CV, moderate positive correlations between Age-CV, CCT-CH, CCT-CRF, ACD-CD, CH-CD. Scatter plots of correlations of age can be found in Figure 1.

In the emmetropic group, there are strong negative correlations between Age-ACD, Age-CD, moderate negative correlations between Age-CH, Age-CRF, AL-CD, Hex-CD, ACD-CV, Hex-CV and strong positive correlations between CCT-CH, CCT-CRF, ACD-CH, ACD-CRF, CRF-CD, CH-CD, moderate positive correlations between Age-CV, ACD-CD, SE-CD, Hex-AL (Table 4).

## 4. Discussion

Corneal biomechanics represent an emerging domain in adult and pediatric ophthalmology, with proven value in refractive surgery [14] and diseases such as glaucoma [15,16], keratoconus [17], and other refractive errors such as myopia [18]. In our study, both CH and CRF were significantly higher in hyperopes compared to emmetropes, and in children compared to adults. A large scale study, involving over 93,000 eyes, has led to similar results: CH is higher in younger people, however that study has involved significantly older people (between the ages of 40 and 69), and has also revealed a difference between genders in terms of CH, which was not present in our study [19]. A study which divided the participants in age decades revealed that CH and CRF are significantly different between the ages of 10 and 69, with the average values in the 10–19 age bracket being most significantly higher than in other decades. Moreover, CH and CRF were on average higher in females and, similar to our study, were higher in hyperopes (compared to myopes and emmetropes) [20].

Biologically, this may be explained as aging induces reduced elasticity and compliance in the cornea through the effect of oxidative stress, protein glycation and, ultimately, collagen crosslinking [21]. 

There is an important relationship between corneal thickness and corneal hysteresis and resistance factor—a strong correlation has been established through multiple studies [22,23], including through multivariate analysis [24]. This can be easily explained, as the elasticity and viscosity of the cornea, of which CH and CRF are markers, are increased as the corneal thickness is increased [25]. 

Corneal thickness is an ocular parameter which may be influenced by several systemic pathological processes—such as accumulation of advanced glycation end products in the stroma or endothelial dysfunction, which all lead to an increase in central or peripheral corneal thickness [26]. Specifically, an increase of corneal thickness has been detected in diabetes mellitus (DM) [27], hyperparathyroidism, gout, and a decrease in connective tissue disease such as Ehlers-Danlos Syndrome, Marfan Syndrome [26].

In our study, CCT is correlated with CH and CRF both in emmetropes and in hyperopes, and the latter two are different between the two refractive groups. However, no significant difference in CCT was observed—it is known that corneal biomechanics are influenced by biological properties such as the tridimensional organization of collagen fibers, extracellular matrix components, or osmotic pressure [28], and thus may explain the increased CH and CRF of the hyperopic cornea, for the same CCT.

One important relationship to discuss pertains to the anterior chamber depth. Hyperopia, a shallow central anterior chamber and a short axial length are all known risk factors for primary angle-closure glaucoma [5,29]. As expected, our study reveals a lower AL and ACD in hyperopic patients. Interestingly, it also reveals a negative correlation between age and ACD, both in emmetropes and hyperopes, suggesting a lower anterior chamber depth as patients grow older. Other studies confirm this association between ACD and either age or refractive error, some data even supporting the fact that the largest rate of ACD decrease occurs in the second decade of life [30]. In tandem with the anterior chamber depth, lens parameters are of importance in angle-closure glaucoma. It is reported that in PACG, the lens thickness is higher and the relative position of the lens is more anterior [31]. 

A Cochrane review suggests that, in PACG, lens extraction, which acts on relieving the pupillary block and on increasing the ACD, is a feasible therapeutic approach, with benefits in terms of visual field progression and quantity of IOP-lowering medication needed [32].

Several studies have investigated the correlation between CH and CRF and morphological parameters, such as the ACD or AL [25,33]. However, results were conflicting—in our study, there is a positive correlation between ACD and the corneal biomechanical parameters in emmetropes, while the correlation is statistically significant between AL and corneal biomechanics in hyperopes. In a study of almost 1000 eyes, linear regression analysis identifies anterior chamber depth and volume as factors influencing CH in a model adjusted for age and gender, however the correlation is no longer significant in a multivariate model which includes other factors such as CCT or corneal curvature [25]. In a study of pediatric eyes, a multivariate analysis reveals that CH and CRF are both negatively correlated with AL, with no correlation with ACD [33]. Similarly, in our hyperopic cohort ACD did not correlate with corneal biomechanics, while in the emmetropic group an inverse correlation was found. These differing results suggest that more research is needed, and that refractive state may have a big influence over the biomechanical—morphological interactions.

The corneal endothelium is the innermost layer of the cornea, consisting of tightly interconnected cells, with an essential role in maintaining proper corneal hydration and transparency [34]. It is a single-cell layer, with limited capacity for regeneration [35]. In normal eyes, the annual rate of cell loss is 0.6%, and there are several systemic and ocular conditions which may increase this rate during the course of the patient’s life [34,36]. Similarly, in our study there is a correlation between age and cell density and variability, also a significant difference between adults and children, which suggest a decrease in density and uniformity of endothelial cells as patients age. However, as the present study is cross-sectional, a rate of cell loss could not be calculated to compare the hyperopic and emmetropic patients. 

The corneal endothelium is an ocular structure which may be influenced by systemic conditions, such as diabetes mellitus. Several morphological alterations have been recorded in DM, including reduced cell density, polymorphism, and a higher cell loss rate which correlates with longer disease duration and low glycemic control [37]. Endothelial cell dysfunction has been described also in the context of hyperlipidemia, smoking or in patients with a history of ischemic stroke [34].

As stated previously, hyperopia is a significant PACG risk factor, and several studies have found lower CD in PACG compared to open-angle glaucoma [38] or compared to healthy controls [35]. Both in emmetropes and in hyperopes we have found a correlation between CD, CV and ACD—a shallower anterior chamber is correlated with a decrease in endothelial cell density and increase in variability, which may have important repercussions over the patient’s lifetime.

Our study has found statistically significant, moderate-to-strong correlations between CH and CRF and endothelial cell count and variability. However, to the best of our knowledge, these findings differ from those in the literature, where no significant correlation has been found between biomechanical and endothelial corneal parameters in healthy volunteers [39] or in patients with cataract [40]. As the level of corneal hydration, which is regulated by the endothelial pump function, may influence corneal biomechanics [39], more studies are needed in order to identify the factors that modulate the relationship between CH, CRF and endothelium parameters.

Our study shows the inverse correlation between age and biomechanical parameters (CH and CRF), anterior segment morphology (ACD) and endothelial layer parameters (Cell density, variability or hexagonality). However, it is cross-sectional; therefore, it does not follow the patients’ evolution over time, in relation to these variables. Prospective follow-up studies of hyperopic cohorts are needed, in order to assess this evolution over decades and to evaluate the PACG risk.

## 5. Conclusions

Our research suggests a significant difference between young emmetropes and hyperopes in terms of ACD, CH and CRF. Moreover, there are significant correlations in the hyperopic group: negative between age and either ACD, CD, CH or CRF, between morphological and biomechanical parameters (AL, CCT and CH, CRF), endothelial and either morphological or biomechanical parameters (ACD-CD, ACD-CV, CH-CD, CRF-CV, CH-CV). The negative correlation between age and ACD, both in emmetropes and hyperopes, suggests a shallower anterior chamber as patients grow older, increasing the risk for PACG. Considering the amblyopia risk in children and the PACG risk later in life that hyperopia raises, a long-term ophthalmological follow-up plan could be a reasonable suggestion for young hyperopes.

## Figures and Tables

**Figure 1 medicina-59-01660-f001:**
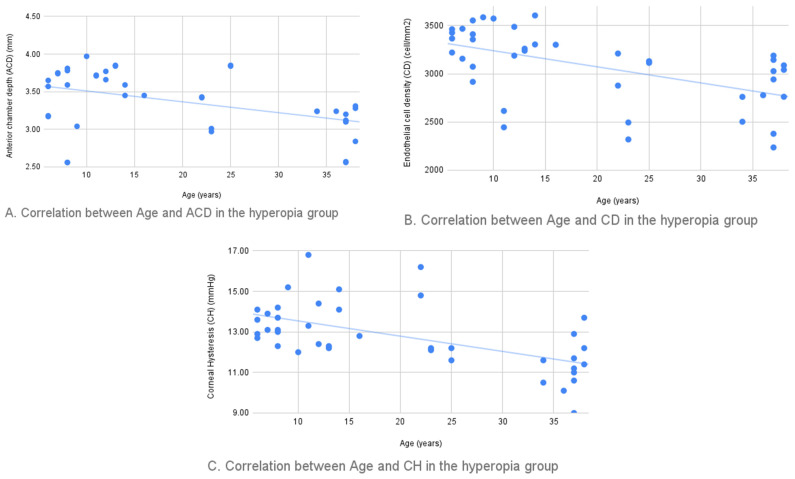
Correlations of age in the hyperopia study group: with anterior chamber depth (scatter plot (**A**)), with endothelial cell density (scatter plot (**B**)), with corneal hysteresis (scatter plot (**C**)).

**Table 1 medicina-59-01660-t001:** Mean and standard deviation of the age, Spherical Equivalent (SE), Axial Length (AL), Anterior Chamber Depth (ACD), Central Corneal Thickness (CCT), Corneal Hysteresis (CH), Corneal Resistance Factor (CRF), Endothelial cell density (CD), Cell variability (CV), Hexagonality (Hex) in the whole cohort and in the hyperopic and emmetropic groups, with Mean Difference, Standard Error and *p* value of Independent Samples *t* Test.

	Mean (±Standard Deviation)				
	Entire Cohort (*n* = 74)	Hyperopic Study Group (*n* = 40)	Emmetropic Control Group (*n* = 34)	Mean Difference (Standard Error)	*p* Value
Age (years)	22.93 (±12.07)	19.83 (±12.37)	26.59 (±10.77)	−6.76 (±2.72)	0.015
SE (D)	1.25 (±1.71)	2.25 (±1.79)	0.07 (±0.29)	2.18 (±0.29)	<0.001
AL (mm)	23.03 (±0.86)	22.54 (±0.68)	23.61 (±0.69)	−1.07 (±0.16)	<0.001
CCT (mm)	0.568 (±0.034)	0.566 (±0.030)	0.569 (±0.038)	−0.003 (±0.008)	0.738
ACD (mm)	3.47 (±0.38)	3.37 (±0.40)	3.59 (±0.33)	−0.22 (±0.09)	0.013
CH (mmHg)	12.39 (±1.56)	12.80 (±1.61)	11.90 (±1.35)	0.91 (±0.35)	0.012
CRF (mmHg)	12.31 (±1.76)	12.71 (±1.91)	11.85 (±1.47)	0.86 (±0.40)	0.036
CD (cells/mm^2^)	3000.65 (±361.51)	3075.70 (±377.27)	2912.35 (±325.63)	163.35 (±82.70)	0.052
CV (%)	28.72 (±4.89)	29.30 (±5.17)	28.03 (±4.52)	0.013 (±0.011)	0.268
Hex (%)	65.68 (±5.26)	65.68 (±5.32)	65.68 (±5.27)	−0.00001 (±0.0123)	0.999

**Table 2 medicina-59-01660-t002:** Mean, standard deviation and *p* value of the Independent Samples *t* Test, regarding the age, Spherical Equivalent (SE), Axial Length (AL), Anterior Chamber Depth (ACD), Central Corneal Thickness (CCT), Corneal Hysteresis (CH), Corneal Resistance Factor (CRF), Endothelial cell density (CD), Cell variability (CV), Hexagonality (Hex) in adult and pediatric subjects, in the hyperopia and emmetropia groups.

	Hyperopic Study Group (*n* = 40)			Emmetropic Control Group (*n* = 34)		
	Adult (*n* = 18)	Children (*n* = 22)	*p* Value	Adult (*n* = 21)	Children (*n* = 13)	*p* Value
Age (years)	32.22 (±6.60)	9.68 (±3.06)	<0.001	34.14 (±5.55)	14.38 (±2.47)	<0.001
SE (D)	2.49 (±2.10)	2.06 (±1.51)	0.458	0.01 (±0.29)	0.17 (±0.26)	0.110
AL (mm)	22.62 (±0.60)	22.48 (±0.74)	0.531	23.55 (±0.59)	23.71 (±0.84)	0.513
CCT (mm)	0.569 (±0.029)	0.564 (±0.032)	0.567	0.561 (±0.032)	0.582 (±0.045)	0.117
ACD (mm)	3.18 (±0.34)	3.52 (±0.39)	0.007	3.39 (±0.21)	3.90 (±0.24)	<0.001
CH (mmHg)	11.94 (±1.69)	13.51 (±1.17)	0.001	11.35 (±1.04)	12.79 (±1.35)	0.001
CRF (mmHg)	11.88 (±2.00)	13.39 (±1.56)	0.011	11.18 (±1.17)	12.93 (±1.28)	0.001
CD (cells/mm^2^)	2833.33 (±321.70)	3274.00 (±298.35)	<0.001	2749.52 (±213.85)	3175.38 (±306.63)	<0.001
CV (%)	31.94 (±4.33)	27.14 (±4.85)	0.002	29.14 (±4.09)	26.23 (±4.75)	0.067
Hex (%)	65.28 (±5.45)	66.00 (±5.31)	0.675	66.86 (±4.39)	63.77 (±6.15)	0.097

**Table 3 medicina-59-01660-t003:** Pearson’s r coefficient and *p* value of correlations between Spherical Equivalent (SE), Axial Length (AL), Anterior Chamber Depth (ACD), Central Corneal Thickness (CCT), Corneal Hysteresis (CH), Corneal Resistance Factor (CRF), Endothelial cell density (CD), Cell variability (CV), Hexagonality (Hex) in the hyperopia group.

Pearson Correlation Coefficients (*p* Value)			
Age-ACD	−0.447 (0.007)	Age-CV	0.470 (0.004)
Age-CH	−0.544 (0.001)	Age-CD	−0.546 (0.001)
Age-CRF	−0.539 (0.001)	ACD-CD	0.509 (0.002)
AL-CH	−0.335 (0.049)	ACD-CV	−0.528 (0.001)
AL-CRF	−0.334 (0.041)	CH-CD	0.384 (0.023)
AL-SE	−0.593 (<0.001)	CRF-CV	−0.438 (0.009)
CCT-CH	0.393 (0.019)	CH-CV	−0.379 (0.025)
CCT-CRF	0.435 (0.009)	CD-CV	−0.396 (0.019)
		Hex-CV	−0.480 (0.004)

**Table 4 medicina-59-01660-t004:** Pearson’s r coefficient and *p* value of correlations between Spherical Equivalent (SE), Axial Length (AL), Anterior Chamber Depth (ACD), Central Corneal Thickness (CCT), Corneal Hysteresis (CH), Corneal Resistance Factor (CRF), Endothelial cell density (CD), Cell variability (CV), Hexagonality (Hex) in the emmetropia group.

Pearson Correlation Coefficients (*p* Value)			
Age-ACD	−0.702 (<0.001)	Age-CV	0.425 (0.014)
Age-CH	−0.455 (0.008)	ACD-CV	−0.390 (0.025)
Age-CRF	−0.445 (0.009)	Hex-CV	−0.463 (0.007)
CCT-CH	0.574 (<0.001)	ACD-CD	0.383 (0.028)
CCT-CRF	0.571 (0.001)	CRF-CD	0.560 (0.001)
ACD-CH	0.566 (0.001)	CH-CD	0.562 (0.001)
ACD-CRF	0.561 (0.001)	SE-CD	0.441 (0.010)
AL-CD	−0.452 (0.008)	Age-CD	−0.553 (0.001)
Hex-CD	−0.372 (0.033)	Hex-AL	0.425 (0.014)

## Data Availability

The datasets generated during and/or analyzed during the current study are available from the corresponding author on reasonable request.

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
