# Peer review of "Particular Anatomy of the Hyperopic Eye and Potential Clinical Implications"

_medicina, 2023, doi:10.3390/medicina59091660_

Round 1

Reviewer 1 Report

1. Grammar and punctuation should be revised.

2. Abbreviations should come after the term and be enclosed in brackets.

3. Some results were stated without standard deviation.

4. The body of the abstract should be revised in terms of punctuation and structure.

5. Where is the IRB (Ethical Committee) approval number and the date of issue?

6. The design of the statistical analysis software program should be mentioned.

7. Standard deviation should be represented by + and –, not + only

8. Figures titles and numbers should be placed under the figure and segregated into A, B, C, etc.

9. References should be updated for the year 2023.

1. Grammar and punctuation should be revised.

Author Response

Thank you for providing valuable feedback on our manuscript Particular anatomy of the hyperopic eye and potential clinical implications submitted to Medicina. We have been able to incorporate changes to reflect the suggestions provided. We have highlighted the changes within the manuscript in blue. Here is a point-by-point response to the comments and concerns:

1. Grammar and punctuation should be revised.

I have revised the manuscript and modified any grammar and punctuation errors.

2. Abbreviations should come after the term and be enclosed in brackets.

I have reviewed the manuscript in this aspect and all modifications are highlighted in blue.

3. Some results were stated without standard deviation.

I have updated the Abstract and Results section and all tables containing such data.

4. The body of the abstract should be revised in terms of punctuation and structure.

I have reviewed the Journal’s guidelines and revised accordingly.

5. Where is the IRB (Ethical Committee) approval number and the date of issue?

The number and date of the approval were inserted after the manuscript, I have updated the Materials and Methods with this information as well.

6. The design of the statistical analysis software program should be mentioned.

The SPSS version 26 was used, as mentioned in Materials and Methods. I have updated with the full denomination of the software program: Statistical Package IBM SPSS Statistics for Windows, version 26 (IBM Corp., Armonk, N.Y., USA).

7. Standard deviation should be represented by + and –, not + only

I have updated the Abstract and Results section with the proper representation ± SD.

8. Figures titles and numbers should be placed under the figure and segregated into A, B, C, etc.

I have updated the subtitles of the 3 charts comprising figure 1 with the correct position and ordering in A, B and C.

9. References should be updated for the year 2023.

I have updated references 26, 27 and 29, now all references have been published in the last 10 years (edited paragraphs in blue - now references [29], [30], [33]).

Reviewer 2 Report

Dear Authors,

The paper “Particular anatomy of the hyperopic eye and potential clinical implications” presents an interesting study regarding the biomechanical and morphological features as well as endothelial parameters of the hyperopic eye. However, I have a few remarks and suggestions.

My suggestions are as follows:

The primary issue regarding the research is the limited number of included patients. Despite this limitation, strong and significant conclusions were derived from the data obtained from small participant groups, which were subsequently subdivided into groups of children and adults.

The exclusion criteria do not encompass diabetes mellitus (DM), a condition known to impact refraction, nor do they consider other systemic diseases that might influence refraction and the number of endothelial cells. It is not specified whether pregnant women are included, despite the recognized impact of pregnancy on eye biomechanics and refraction.

It would also be valuable to determine whether all measurements were conducted under cycloplegia, given its potential influence on anterior chamber depth. Additionally, providing information about the lens thickness would enhance the discussion, considering its effect on anterior chamber depth.

The research did not present data regarding visual acuity and intraocular pressure (IOP) values. Data regarding IOP is significant considering the potential correlation between hypermetropia and glaucoma. Thus, it is recommended to include this information. Moreover, the age range of the participants remains unspecified, as does the upper age limit defining children.

Further, it would be valuable to present the results of comparing all the investigated parameters between adults with hyperopia and their corresponding control group, as well as between children with hyperopia and their respective control group.

In the text within the Discussion section states: “Several studies have investigated the correlation between CH and CRF and morphological parameters, such as the ACD or AL. However, results were conflicting...." It is necessary to provide the references of those studies.

The discussion should be expanded and improved by incorporating more findings from previous research. Furthermore, the authors should discuss the impact of DM and other systemic diseases and conditions on both refraction and the other investigated parameters. Additionally, it’s essential to incorporate data concerning factors that might potentially influence central corneal thickness (CCT), as well as the density of endothelial cells.

Finally, the manuscript should be extended to include more research-related data and references within the introduction and discussion sections.

Author Response

Thank you for providing valuable feedback on our manuscript Particular anatomy of the hyperopic eye and potential clinical implications to Medicina. We have been able to incorporate changes to reflect the suggestions provided. We have highlighted the changes within the manuscript in green. Here is a point-by-point response to the comments and concerns:

  • The primary issue regarding the research is the limited number of included patients. Despite this limitation, strong and significant conclusions were derived from the data obtained from small participant groups, which were subsequently subdivided into groups of children and adults.

Thank you very much for this observation, we obtained strong results and we are looking forward to add more patients in order to confirm these data.

  • The exclusion criteria do not encompass diabetes mellitus (DM), a condition known to impact refraction, nor do they consider other systemic diseases that might influence refraction and the number of endothelial cells. It is not specified whether pregnant women are included, despite the recognized impact of pregnancy on eye biomechanics and refraction.

Thank you for this observation, indeed we have excluded any systemic conditions during the patient’s interview before the examination, but we only noted in the exclusion criteria the ophthalmologic conditions. We updated the list following your request: Furthermore, patients were excluded in the absence of testing compliance (such as low waveform in Ocular Response Analyser testing, under 7), if the patient was pregnant or if they disclosed any systemic pathology (diabetes mellitus, arterial hypertension, dyslipidemia) or systemic chronic medication.

  • It would also be valuable to determine whether all measurements were conducted under cycloplegia, given its potential influence on anterior chamber depth. Additionally, providing information about the lens thickness would enhance the discussion, considering its effect on anterior chamber depth.

All measurements were performed following the complete ophthalmological examination, under cycloplegic conditions. I have clarified this aspect in the Materials and methods sections and highlighted in green. Indeed we have not considered lens thickness as a variable in this cohort, but we intend to include it in future research. Thank you very much for this suggestion. I have expanded the discussion section with studies regarding the lens thickness: 

Other studies confirm this association between ACD and either age or refractive error, some data even supporting the fact that the largest rate of ACD decrease occurs in the second decade of life [30]. In tandem with the anterior chamber depth, lens parameters are of importance in angle-closure glaucoma. It is reported that in PACG, the lens thickness is higher and the relative position of the lens is more anterior [31].

  • The research did not present data regarding visual acuity and intraocular pressure (IOP) values. Data regarding IOP is significant considering the potential correlation between hypermetropia and glaucoma. Thus, it is recommended to include this information. Moreover, the age range of the participants remains unspecified, as does the upper age limit defining children.

Following the ophthalmological examination all patients underwent, and the application of inclusion and exclusion criteria, patients with any ophthalmological disorders (apart from the refractive error hyperopia) were excluded (myopia, keratoconus, amblyopia, cataract, glaucoma, vitreoretinal pathology, presbyopia, history of refractive surgery). Therefore, all patients had IOP in the normal range (11-21 mmHg) and a best corrected visual acuity of 20/20 and we have not included these variables in the analysis. However, we have recognised the role of IOP in the connection between hyperopia and glaucoma, therefore all correlations have been performed with corrections for IOP as a confounder. All paragraphs highlighted in green: 

  • The exclusion criteria were represented by the presence of ocular pathology, other than hyperopia (myopia, keratoconus, amblyopia, cataract, glaucoma, vitreoretinal pathology), the diagnosis of presbyopia or a history of refractive surgery
  • As IOPg may act as a confounding variable, correlations were calculated controlling for it.

I have added this specification in Materials and Methods: Furthermore, patients were included in the pediatric group (age under or equal to 18 years old) and the adult group (age over 18 years old).

I have added the age range of the participants in Results: Average age in the hyperopic group was 19.83 years old (age range 6-38), and in the emmetropic group was 26.59 (age range 10-40).

  • Further, it would be valuable to present the results of comparing all the investigated parameters between adults with hyperopia and their corresponding control group, as well as between children with hyperopia and their respective control group.

Indeed it would be valuable, however in our current patients group the subgroups would have been too small to be relevant. Thank you very much for this suggestion, we are already adding more patients and will include these data in the following papers.

  • In the text within the Discussion section states: “Several studies have investigated the correlation between CH and CRF and morphological parameters, such as the ACD or AL. However, results were conflicting...." It is necessary to provide the references of those studies.

Thank you, I have clarified which references I am referring to ([25][33]), highlighted in green.

  • The discussion should be expanded and improved by incorporating more findings from previous research. Furthermore, the authors should discuss the impact of DM and other systemic diseases and conditions on both refraction and the other investigated parameters. Additionally, it’s essential to incorporate data concerning factors that might potentially influence central corneal thickness (CCT), as well as the density of endothelial cells. Finally, the manuscript should be extended to include more research-related data and references within the introduction and discussion sections.

We have extended the discussion section with issues regarding impact of systemic disease, factors influencing the parameters of the study and the potential of other parameters for future study (such as lens thickness):

Corneal thickness is an ocular parameter which may be influenced by several systemic pathological processes - such as accumulation of advanced glycation end products in the stroma or endothelial dysfunction, which all lead to an increase in central or peripheral corneal thickness [26]. Specifically, an increase of corneal thickness has been detected in diabetes mellitus (DM) [27]gout, and a decrease in connective tissue disease such as Ehlers-Danlos Syndrome, Marfan Syndrome [26].

A study which divided the participants in age decades revealed that CH and CRF are significantly different between the ages of 10 and 69, with the average values in the 10-19 age bracket being most significantly higher than in other decades. Moreover, CH and CRF were on average higher in females and, similar to our study, were higher in hyperopes (compared to myopes and emmetropes) [20].

The corneal endothelium is an ocular structure which may be influenced by systemic conditions, such as diabetes mellitus (DM). Several morphological alterations have been recorded in DM, including reduced cell density, polymorphism, and a higher cell loss rate which correlates with longer disease duration and low glycemic control [34]. Endothelial cell dysfunction has been described also in the context of hyperlipidemia, smoking or in patients with a history of ischemic stroke [31].

[...]Other studies confirm this association between ACD and either age or refractive error, some data even supporting the fact that the largest rate of ACD decrease occurs in the second decade of life [30]. In tandem with the anterior chamber depth, lens parameters are of importance in angle-closure glaucoma. It is reported that in PACG, the lens thickness is higher and the relative position of the lens is more anterior [31].

Sincerely,

Dana Dascalescu M.D.

Reviewer 3 Report

About 10 patients are over the age of 35. Presbyopia might play a role and bias your results.

Age influences the endothelial count as well and might be a bias too.

The changes in the AC depth are a known fact in hyperopia.

Though interesting - how can your findings influence our daily clinical routine?

Please address these issues.

Author Response

Thank you for providing valuable feedback on our manuscript Particular anatomy of the hyperopic eye and potential clinical implications submitted to Medicina. We have been able to incorporate changes to reflect the suggestions provided. We have highlighted the changes within the manuscript in orange. Here is a point-by-point response to the comments and concerns:

  • About 10 patients are over the age of 35. Presbyopia might play a role and bias your results.

This confounder has been considered while planning the research, and a diagnosis of presbyopia (as resulted in the complete ophthalmological examination) is an exclusion criteria. I have highlighted it in the Materials and methods section in orange: The exclusion criteria were represented by [...], the diagnosis of presbyopia or a history of refractive surgery.

  • Age influences the endothelial count as well and might be a bias too.

Indeed, we found significant correlations between endothelial parameters and age, however, as the present study is cross-sectional, a rate of cell loss could not be calculated to compare the hyperopic and emmetropic patients. I have expanded the literature review regarding systemic factors influencing endothelial cell loss rate: 

In normal eyes, the annual rate of cell loss is 0.6%, and there are several systemic and ocular conditions which may increase this rate during the course of the patient’s life [34,36]. Similarly, in our study there is a correlation between age and cell density and variability, also a significant difference between adults and children, which suggest a decrease in density and uniformity of endothelial cells as patients age. However, as the present study is cross-sectional, a rate of cell loss could not be calculated to compare the hyperopic and emmetropic patients. 

The corneal endothelium is an ocular structure which may be influenced by systemic conditions, such as diabetes mellitus. Several morphological alterations have been recorded in DM, including reduced cell density, polymorphism, and a higher cell loss rate which correlates with longer disease duration and low glycemic control [37]. Endothelial cell dysfunction has been described also in the context of hyperlipidemia, smoking or in patients with a history of ischemic stroke [34].

  • The changes in the AC depth are a known fact in hyperopia.

Indeed, a shallow AC is a known element in hyperopia. Our research builds upon this known fact and reveals an interesting connection between age and AC depth, which suggests a decrease of the AC as hyperopic patients age. We consider this of clinical value, as we may encourage our patients to maintain a schedule of ophthalmological consultations as they age, focusing on parameters related to angle closure.

  • Though interesting - how can your findings influence our daily clinical routine?

Our study supports the idea that certain transformations related to disease (decrease in endothelial cell number, anterior chamber depth, corneal hysteresis) may correlate with age at younger ages than is usually believed (children and young adults in our case). Thus, we support suggesting to our young hyperopic patients to upkeep a schedule of ophthalmological consultations, as stated in the conclusions: Considering the amblyopia risk in children and the PACG risk later in life that hyperopia raises, a long-term ophthalmological follow-up plan could be a reasonable suggestion for young hyperopes.

Sincerely,

Dana Dascalescu M.D.

Round 2

Reviewer 2 Report

The manuscript “Particular anatomy of the hyperopic eye and potential clinical implications” presents an interesting study regarding the biomechanical and morphological features of the hyperopic eye. It offers a valuable foundation for future research and practical applications. The authors have adequately addressed and incorporated the provided suggestions, and the manuscript could be accepted in present form.

Reviewer 3 Report

It is good now